# Can Circadian Eating Pattern Adjustments Reduce Risk or Prevent Development of T2D?

**DOI:** 10.3390/nu15071762

**Published:** 2023-04-04

**Authors:** Carlee Harris, Krzysztof Czaja

**Affiliations:** Department of Biomedical Sciences, College of Veterinary Medicine, University of Georgia, Athens, GA 30602, USA

**Keywords:** eating patterns, type 2 diabetes, microbiome, intermittent fasting, long-term fasting, time-restricted eating, alternate-day fasting, 5:2 fasting, circadian rhythm, metabolic hormones

## Abstract

Type 2 diabetes (T2D) is a chronic condition that occurs in insulin-resistant people with reduced glucose uptake. It is contributed to and exacerbated by a poor diet that results in accumulation of adipose tissue, high blood sugar, and other metabolic issues. Because humans have undergone food scarcity throughout history, our species has adapted a fat reserve genotype. This adaptation is no longer beneficial, as eating at a higher frequency than that of our ancestors has had a significant effect on T2D development. Eating at high frequencies disrupts the circadian clock, the circadian rhythm, and the composition of the gut microbiome, as well as hormone secretion and sensitivity. The current literature suggests an improved diet requires meal consistency, avoiding late-night eating, low meal frequency, and fasting to increase metabolic health. In addition, fasting as a treatment for T2D must be used correctly for beneficial results. Early time-restricted eating (TRE) provides many benefits such as improving insulin resistance, cognitive function, and glycemic control. Alternate-day fasting (ADF), 5:2 fasting, and long-term fasting all have benefits; however, they may be less advantageous than early TRE. Therefore, eating pattern adjustments can be used to reduce T2D if used correctly.

## 1. Introduction

Diabetes is a multisystem disease that results from deficiencies in insulin secretion and insulin resistance [1,2,3]. Onset of type 1 diabetes usually occurs during childhood and is caused by low levels of insulin secretion, while T2D usually occurs during adulthood and in most cases is caused by the inability of the pancreas to produce sufficient insulin combined with deficiencies in the body’s response to insulin. T2D, the fourth leading cause of death worldwide, is associated with obesity and poor diet [4]. Diabetic insulin resistance is caused by post-binding defects, and impaired beta-cell function advances these issues by altering insulin secretion [2,5]. T2D occurs when early insulin secretion decreases, and the pancreas is unable to overcome insulin resistance [2,3]. Insulin resistance and decreased insulin secretion cause hyperglycemia and many other issues related to obesity that worsen the state of the disorder such as elevated free fatty acids and ectopic fat accumulation [2,6]. This chapter will provide insight into the mechanistic properties of T2D.

Some studies show that high sugar and fat intake are associated with T2D risk [4]. However, a recent human study suggests that some forms of dietary saturated fatty acids, such as those found in cheese, may be associated with reduced T2D risk [7]. Moreover, a study shows that body mass index (BMI) and weight gain have a positive relationship with T2D risk in people, suggesting that it is largely caused by poor dietary practices [8]. Adipose tissue secretes non-esterified fatty acids and hormones that affect metabolic processes and lead to insulin resistance, and the development of obesity results in large amounts of adipose tissue and an increased secretion of these substances [9,10]. These findings show the importance of the role of diet on T2D and suggest that diet management may be useful for both preventing and treating the disease.

Insulin, a hormone released by the pancreas, signals glucose uptake, and if there are complications during the response, insulin resistance occurs. A few studies on humans with T2D conclude that insulin resistance is characterized by low muscular glucose uptake and inefficient suppression of hepatic glucose production [11,12]. Abnormal glucose uptake is generated by abnormalities that occur after insulin binds to insulin receptors [5]. The results of a human study suggest that individuals with T2D have under-expressed and impaired glucose transporters that prevent muscular tissue from glucose uptake [13]. Defects in intracellular insulin signaling also contribute to abnormal glucose uptake [2]. A human study suggests that not only is glucose uptake decreased, but also once glucose enters the cell, glucose phosphorylation occurs at a slower rate [14]. Glucose phosphorylation is a crucial process in glucose metabolism, and the complications of this step result in excess free glucose within cells [5]. Post-binding mutations also impair glycogen synthesis and other metabolic effects of insulin [5]. As a result, in those with T2D, impaired glucose transporters, intracellular insulin signaling, and glucose phosphorylation may occur within muscle cells, resulting in reduced glucose uptake.

The precursors of T2D can be identified by an overproduction of insulin in an attempt to maintain normal glucose levels despite insulin resistance, as seen in a study on normal and obese people [6,15]. A study on rats indicates that the increase in beta-cell mass alters beta-cell enzymes, decreasing beta-cell function, altering insulin secretion, and advancing glucose uptake mechanistic abnormalities [6,16]. T2D develops when early-phase insulin secretion can no longer overcome insulin resistance, and postprandial hyperglycemia occurs [2,3]. Abnormal first-phase insulin secretion can be caused by exposure to diabetic environments in utero or by the effects of weight gain [3]. The reduction in insulin secretion occurs because people suffering from diabetes and obesity have fewer incretin hormones that increase insulin secretion [17]. Early-phase insulin is responsible for shifting the body from a fasting to a prandial state and preparing the body for glucose uptake [3]. Because they have insulin resistance, lower early-phase insulin secretion prevents those with type 2 diabetes from being able to perform the metabolic mechanisms necessary for glucose uptake, further contributing to hyperglycemia and T2D.

Reduced glucose uptake results in the release of excess free fatty acids [2,6]. A human study suggests that high levels of free fatty acids further inhibit glucose metabolism and insulin sensitivity [6,18]. Increased levels of free fatty acids lead to increased ectopic fat accumulation and reduced beta-cell function, which both advance glucose uptake deficiencies [2,6]. Excessive free fatty acids are also responsible for impairing intracellular signaling in humans [19]. In people with T2D, the activity of insulin receptors—a type of tyrosine kinase receptor—is reduced, not because of structural abnormalities or insulin binding affinity, but because of the metabolic disturbances present [5,20]. Reduced glucose uptake leads to the elevated secretion of free fatty acids, increased ectopic fat accumulation, reduced beta-cell function, impaired intracellular signaling, and reduced insulin receptor activity. These factors then combine to advance insulin resistance.

The understanding of T2D, a disease originally thought to only involve abnormalities in the pancreas, has improved over the years [1]. T2D is now known to be a multisystem disorder involving the pancreas, muscle tissue, fat cells, kidneys, and brain [1]. The appreciation of the complexity of the mechanisms of T2D assists in the development of more effective treatments and management strategies [1]. For example, growing evidence suggests that restoring the timing and amount of early-phase insulin secretion could be used as a therapy for improving glucose tolerance, and thus reducing T2D in humans [21]. However, T2D cannot be treated with pharmacotherapy alone. Human studies discuss the benefits of weight-related prevention and management of chronic diseases related to obesity, such as T2D [8]. The disease must be treated with a combination of therapies and improvements in lifestyle [1]. Hence, T2D patients should be well educated on dietary management strategies to properly treat the disease [4]. As the understanding of T2D mechanisms has advanced, treatment is possible when informed patients utilize a combination of therapies and improvements in lifestyle. This review will provide insight into potential ways to reduce T2D prevalence using adjustments in eating patterns in light of the known mechanisms of T2D. The collective available data include both human and animal studies, and conflicting arguments may be observed due to anatomical differences.

## 2. Modern Eating Patterns: The Cause of Type 2 Diabetes

The modern non-3rd world human eating pattern of three meals a day with intermittent snacking is far from that of our ancestors and may be largely responsible for the elevated rates of chronic diseases such as obesity and T2D. Ancestral humans adapted to harsh conditions such as famine, infectious plagues, and cold weather, which resulted in a genetic adaptation that allowed for the storage of excess energy as fat [22,23]. Studies suggest that this one-time evolutionary advantage may now contribute to the current obesity epidemic by encouraging fat storage, which is no longer essential [23]. This chapter will consider the change in eating patterns seen throughout history, as well as the effects these adjustments have on the T2D epidemic.

### 2.1. History of Eating Patterns

Currently, the most common eating pattern, three meals and snacks every day, is caused by around the clock access to food due to urbanization and is a recent change in behavior [24,25]. Throughout human history, most ate less frequently than modern humans. For example, ancient Romans ate one meal each day with two smaller snacks and believed more than one meal a day was unhealthy [24]. Moreover, preindustrial and paleolithic societies periodically underwent severe shortages and were forced to eat less often [22]. However, over time, meals began to serve other purposes. During medieval times, monasticism influenced the common person’s eating patterns [24]. The practice of having mealtimes devoted to prayer and religion established breakfast as an important mealtime, as it allowed for the first prayer of the day [24]. Monks were also required to remain silent during meals, to allow others present to read religious texts aloud [24]. This tradition continued into the industrial revolution when people ate a meal before work [24]. However, some studies suggest that our agrarian ancestors were the first to utilize a three-meal eating pattern [25]. This is because the agricultural revolution led to a year-round supply of food [25]. Further, the common late dinner time is a recent change due to artificial light, as it allows for eating after dark [24]. For example, the largest meal of the day, dinner, was consumed around noon in medieval England [26]. Dinner times were pushed back during the industrial revolution, as the working class was occupied during the day [26]. Therefore, lunch was introduced to break the extended fasting period between breakfast and dinner [26]. Thus, the modern eating pattern has not been seen throughout most of human history, as most ate fewer meals, and the last meal of the day was pushed back due to work schedules and artificial light.

There is a mismatch between human evolution and the modern lifestyle, as this once essential and favorable genotype predisposes obesity and T2D [23,27] This mismatch occurs because our genetics have minimally changed since the appearance of modern humans roughly 40,000 years ago, and industrialization has changed our diet as well as our activity patterns [22,27]. This is because the industrial revolution, agribusiness, and modern food processing techniques were developed too recently to have an evolutionary effect [22]. A review discusses that ancestral humans favored efficient energy extraction and storage [28]. While modern humans still have this trait, it is now not needed in industrialized settings with a surplus of easily accessible and calorie-dense foods [28]. This surplus alongside the modern nonactive lifestyle leads to an energy imbalance [28]. As a result, the fat reserve genotype adapted by our ancestors is no longer needed in today’s society, has led to a high incidence of obesity, and is related to the growing prevalence of T2D.

### 2.2. High vs. Low Meal Frequency

Many studies compare the effects of modern and traditional eating patterns, and the results suggest that the common modern three daily meals frequency has negative effects. Many human studies show that eating more than three meals a day was associated with an increase in T2D risk, while eating less than three meals a day was associated with a decrease in T2D risk [29,30,31]. In addition, those with obesity have more frequent snacking and high-energy intake than people without obesity [32]. These relationships were also seen during the recent COVID-19 pandemic, when obesity increased along with snacking and meal frequency [33]. However, there is some concern about skipping meals, as seen in a study on adults [34]. This study shows that while meal skipping does reduce net daily energy intake, diet quality is also reduced, which could have negative long-term effects [34]. The observed change in diet quality occurs because breakfast, lunch, and dinner usually include foods contained in a healthy diet, which are not generally consumed outside of meals [34]. Out of all three meals, skipping dinner had both the largest impact on reducing caloric intake and the lowest impact on diet quality, and skipping breakfast reduced diet quality the most [34]. Moreover, skipping breakfast increases T2D risk, as it reduces insulin sensitivity and metabolic health [35]. This is supported by a study that found irregular breakfast consumption and T2D risk to have a positive association in women [36]. Though there is a positive association between meal and snack frequency and T2D risk, skipping meals should be used with caution.

To conclude, modern T2D incidence has been highly influenced by the recent human deviation from ancestral eating frequencies in conjunction with the fat reserve genotype [22,23]. The high meal frequency seen in common modern eating patterns causes metabolic issues such as T2D, as humans are adapted to intermittent energy intake [25]. Therefore, reducing meal frequency may be used to decrease T2D risk; however, meal skipping must be used with caution because of the impact on diet quality [34].

## 3. The Circadian Clock, The Gut Microbiome, and Metabolic Hormones

The circadian 24 h clock is responsible for the natural metabolic processes the body undergoes, including maintaining metabolic homeostasis through enzyme regulation [25,37]. Interestingly, the circadian clock is influenced by light exposure and time of feeding in order to synchronize circadian rhythms to the environment [38]. The central circadian clock is located in the suprachiasmatic nuclei on the anterior hypothalamus of the brain, and peripheral clocks are found in many other parts of the body such as the liver, intestine, and retina [37]. The mechanism of the circadian clock begins when the suprachiasmatic nuclei receive information and transmit it to peripheral clocks through the circulatory or nervous systems [37]. Circadian clocks are also composed of oscillator cells, which are then responsible for sending out the circadian rhythms [37]. This chapter considers how current eating patterns lead to disrupted circadian rhythm and the many issues that stem from this disruption such as negative changes to the gut microbiome, increased hormone resistance, and T2D.

### 3.1. The Gut Microbiome

The gut microbiome may be the link between inflammation and insulin resistance seen in T2D, as seen in a human study [39]. The gut microbiome refers to the composition of gut bacteria, viruses, fungi, archaea, and phages [39]. A rodent study suggests that disrupted circadian rhythms in the host influence the bacterial populations of the intestine [40]. In addition, changes in both quantity and diversity of gut microbiota can lead to the progression of metabolic disorders, as seen in non-obese diabetic mice [41]. One of the roles of microbiota, as seen in a human fecal metagenome study, is to maintain intestinal homeostasis of the gut epithelial lining [42]. Reduction in microbiota diversity breaks the cell-to-cell integrity, leading to a leaky gut with increased permeability and intestinal inflammation [43]. Therefore, dysbiosis of gut microbiota has been linked to obesity and T2D, as it reshapes intestinal barrier functions and signaling pathways [43]. The negative effects of disrupted circadian rhythm are supported by a study on the intestinal microbiota of jet-lagged, circadian-disrupted humans [44]. When the microbiota was transferred to mice, the mice had increased levels of obesity and glucose intolerance [44]. Changes in dietary patterns can alter the Firmicutes/Bacteroidetes ratio, with increased Bacteroidetes leading to increased energy yield [43]. A recent report suggests that the microbiota of people with obesity is more efficient at extracting energy from their diet when compared to those without obesity [39]. This could in part explain the high levels of fat accumulation and storage seen in those with obesity. in addition, the abundance of Firmicutes relative to Bacteroidetes is positively correlated with the lean tissue index in people with T2D [45]. Lastly, subjects with T2D tend to have a lower Firmicutes/Bacteroidetes ratio than normal individuals [45]. Therefore, increasing the Firmicutes/Bacteroidetes ratio may be beneficial in those with T2D. As a result, disruption in the circadian rhythm of the host may disrupt gut microbiome composition, leading to the increased insulin resistance and buildup of excess energy seen in T2D and obesity.

### 3.2. Effects of Modern Eating Patterns on Hormone Secretion and Sensitivity

Many hormones such as insulin, leptin, and ghrelin show circadian oscillation, as the circadian clock is responsible for the patterns seen with these hormones [37]. For example, as seen in a rat study, ghrelin is a hormone responsible for appetite regulation, and ghrelin secretion occurs before feeding and falls within one hour after eating [46]. Studies that examined ghrelin levels during fasting support the idea that ghrelin levels are influenced by the circadian clock. During fasting, subjects experienced a rise in ghrelin levels around mealtimes, which spontaneously decreased within two hours without eating [47]. Moreover, a low-frequency diet has beneficial effects on decreasing ghrelin levels, as seen in a study on healthy males [48]. Those with a low meal frequency diet had ghrelin levels that dropped lower than 35 pg/mL in the serum within the first hour after eating, while those practicing a high-frequency diet had ghrelin levels of about 45 pg/mL one hour after eating [48]. This relationship continues throughout the day, as those with high meal frequency have higher levels of ghrelin throughout the day [48]. This suggests that low meal frequency improves the ghrelin hormone response, as the circadian clock is responsible for metabolic hormone secretion.

The observed relationship between eating frequency and T2D risk also occurs because eating many times a day causes mildly elevated blood glucose and insulin levels, with lower peaks and troughs in women [31]. Glucose levels in men with low meal frequency stretched from values lower than 4 mmol/L to as high as 7 mmol/L, while those with high meal frequencies had glucose levels from about 5 to 6 mmol/L [48]. This trend was also seen in insulin, as those with low meal frequencies had insulin levels ranging from about 10 uU/mL up to 70 uU/mL, while those with high meal frequencies had insulin levels mostly within a range of about 30 to 45 uU/mL [48]. This suggests that a low meal frequency allows for an increased hormonal response. In addition, a human study suggests that constant elevated insulin and glucose levels seen in high eating frequencies require continuous insulin secretion, putting stress on the pancreas and preventing the release of counterregulatory hormones [31]. The response of leptin to meal frequency also explains the observed relationship between meal frequency and T2D. Leptin is a hormone responsible for regulating food intake, as leptin binds to receptors that then send signals to inhibit food intake [49]. Leptin secretion is regulated by food intake, total body fat, and other hormones such as insulin [49]. For example, prolonged hyperinsulinemia leads to an increase in leptin secretion [49]. Therefore, as chronic overfeeding alters leptin levels, it may also lead to leptin resistance, causing metabolic complications [49]. Further, studies on people suggest that leptin levels decrease during fasting, but increase during overfeeding [50,51]. A positive relationship exists between serum leptin levels and the percentage of body fat, as leptin is overproduced in those with obesity [49]. This was supported by a study comparing leptin levels during fasting in people with obesity to those without [50]. It was found that those with obesity began with leptin levels higher than 31 ng/mL, and after 24 h of fasting, levels decreased to around 12 ng/mL [50]. In contrast, those without obesity began with leptin levels around 11 ng/mL, and after 24 h of fasting, the levels decreased to around 4 ng/mL [50]. This suggests that periodic fasting may be beneficial to those with obesity, in terms of reducing leptin levels. Therefore, chronic overfeeding may lead to leptin and insulin resistance, as high-frequency eating patterns lead to increased hormone secretion.

### 3.3. The Effects of Urbanization on Type 2 Diabetes

Urbanization is responsible for the increase in diabetes and obesity for many reasons beyond increasing meal frequency and decreasing famine. An effect of urbanization is irregular meal routines, such as skipping breakfast and eating at irregular mealtimes, disrupting circadian rhythms. A study on human meal irregularity suggests that an irregular meal routine disrupts the circadian clock and causes an increase in cardiometabolic risk [52]. Additionally, adults with inconsistent eating patterns have a higher BMI than those with consistent meal routines [53]. These relationships could be explained by the effect of circadian misalignment on leptin levels in mice with light-phase-restricted eating [54]. As the circadian clock is related to leptin levels, overeating can be stimulated during abnormal mealtimes in mice [54,55]. The negative effects of circadian misalignment in humans may also be explained by a study on genetically modified mice with mutations in the circadian rhythm genes [56]. This study shows that genetically modified mice have an increased risk of low glucose tolerance [56]. Mutated mice were at increased risk of developing T2D and obesity, indicating the importance of the circadian clock network [56]. Furthermore, a study on people with prediabetes who completed oral glucose tolerance tests in the morning and evening shows that glucose tolerance decreases over the day, as evening meals result in a more hyperglycemic response than identical morning meals [57]. This has been related to an increase in insulin resistance that occurs over the course of a day, as found in a study on healthy men [58]. These findings, therefore, suggest that late eating may have a negative impact on those with T2D. As a result, eating at inconsistent mealtimes and late-night eating are both examples of eating patterns associated with the effects of urbanization that are contributing to the T2D epidemic.

### 3.4. Reducing the Effects of Urbanization

Eating pattern adjustments can be made to reduce some of the effects of urbanization on the T2D epidemic. Because of the adaptations made by ancestral humans, utilizing a diet more similar to our ancestors may cause a decline in T2D [22,23]. Because of the negative effects associated with leptin levels and circadian misalignment, eating on regular meal routines could help decrease T2D risk [55]. Additionally, earlier mealtimes are associated with weight loss for obese subjects because they extend daily fasting time [25,36]. Therefore, avoiding late eating, and thus providing a longer overnight fast, is an eating pattern more similar to that of our ancestors, as we are adapted to intermittent energy intake [25]. Moreover, eating breakfast has benefits for metabolic health and could prevent the onset of T2D in some individuals [35]. However, some studies suggest that having breakfast as the largest meal of the day is associated with the greatest benefits, while others believe lunch is the better time to eat the largest meal [29,59]. Therefore, reducing dinner may be the best approach to reducing T2D, as a longer overnight fast is more similar to the eating pattern of our ancestors, and earlier eating times seem to reduce T2D risk.

In conclusion, many aspects of the common eating pattern lead to circadian misalignment, resulting in many metabolic issues such as disrupted gut microbiome and altered hormone sensitivity [6,31,40,44,51]. Current eating patterns that increase T2D risk include overfeeding, high meal frequency, irregular meal routine, skipping breakfast, and late eating. Eating patterns that reduce T2D include regular meal routines, eating breakfast, and decreasing meal frequency when the effects on diet quality are reduced. As mentioned previously, skipping dinner has the smallest effect on food quality and the largest effect on caloric reduction [34]. Further, reducing late eating for a longer overnight fast has beneficial results [25]. Therefore, a consistent meal routine with breakfast and lunch may be the best approach to reducing T2D.

## 4. Is Fasting a Treatment for Type 2 Diabetes?

Fasting is the voluntary absence or limitation of caloric ingestion for a period of time, causing changes in the activity of numerous signaling pathways [60]. Fasting has been used for religious purposes throughout history, as seen during Ramadan, a holy month of fasting for Muslims [61]. Christians, Jews, Buddhists, and Hindus also fast on designated days of the week or year [62]. Fasting can be classified as intermittent or long-term. Intermittent fasting occurs for less than two days and includes ADF, TRE, and 5:2 fasting. Intermittent fasting includes a variety of eating and fasting windows, which have different outcomes in those with T2D. These fasting periods have windows ranging from 12 to 24 h and can be categorized as TRE or ADF. During TRE, fasting occurs for a portion of the day, allowing for food intake within a window of the day, and during ADF, normal diet occurs one day and restricted food intake occurs the next [60]. Five:two fasting is a form of fasting where caloric intake is restricted for two nonconsecutive days of the week, with sensible, but nonrestrictive eating on the remaining days. Another type of fasting, known as long-term fasting, is the process of fasting for two to twenty-one days [60]. This chapter will discuss the different effects of each fasting type, as well as provide insight into which methods could best reduce the prevalence of T2D.

### 4.1. Time-Restricted Eating

Because the time of day when eating occurs affects body weight, body composition, glucose regulation, and overall health, there are different effects of fasting at different points of the day [63]. For example, human studies suggest that early TRE reduces fasting glucose levels, while late TRE increases fasting glucose levels [64,65]. However, the results of a recent study on men with prediabetes suggest that early TRE improved insulin sensitivity and beta-cell responsiveness but did not improve blood glucose levels [66]. This suggests that early TRE may be effective at improving insulin sensitivity without directly reducing glucose levels [66]. In addition, earlier TRE leads to beneficial outcomes because of improved circadian glucose homeostasis, reduced oxidative stress, improved beta cell function, increased flux, and increased ketone body production [63]. The effects of late TRE can be observed from studies on Ramadan, a holy month of fasting that occurs once a year where Muslims fast from dawn until sunset. A study on those with T2D participating in Ramadan reveals that many parameters such as HbA1c, glucose levels, BMI, and weight were not significantly altered [61]. Therefore, T2D can still occur in those participating in Ramadan, suggesting that late TRE has few benefits. Furthermore, a meta-analysis discussing twelve randomized controlled human trials that compared the effects of early and late TRE found that current data suggest that early TRE is more effective at providing metabolic benefits than late TRE [67]. Specifically, while both early and late TRE lead to weight loss and glycemic metabolic benefits, early TRE had a more significant effect on improving HOMA-IR [67]. As a result, in many cases, early TRE has some superior effects over late TRE in terms of reducing T2D parameters.

The effect of eating window size in TRE has been examined. A study on individuals with obesity compared 12 h fasting and 14 h fasting within a day during TRE [68]. Those who fast for 12 h have a large, 12 h eating window, while those who fast for 14 h have a smaller, 10 h eating window [68]. Those who participated in 14 h fasting lost 1.9 kg more weight than those who used 12 h fasting [68]. Additionally, the 14 h fasting group showed a reduction of 8 mg/dl in fasting blood glucose, while the fasting blood glucose of the 12 h fasting group was only reduced by 3 mg/dl [68]. While 14 h fasting appears to be more successful at improving diabetic parameters, these results are statistically not strongly significant [68]. This suggests that reducing eating window size may not be beneficial for those with T2D. However, others discuss that window size of TRE may impact caloric intake. Researchers collected data on the feasibility of TRE with a 6 h eating window, and they found that many experienced more difficulty eating within a 6 h window than fasting for 18 h [66]. This is supported by a study that tested 4 and 6 h TRE in adults with obesity [69]. The results suggested improvements in cardiometabolic health as intermittent fasting reduces energy intake by 550 kcal per day without the stress of calorie counting [69]. Another study on overweight humans found that those undergoing 8 h TRE had fewer eating occasions than those who did not undergo TRE [70]. This suggests that TRE with small eating windows may reduce caloric intake due to the difficulty of consuming the same number of calories within a smaller amount of time. Thus, while some studies show insignificant results for reducing the eating window in TRE, others show evidence supporting the benefits of reducing the eating window due to its effect on caloric intake.

Many of the beneficial metabolic effects of early TRE may be explained by TRE’s influence on the circadian system in overweight adults [65]. This is because eating in sync with the circadian rhythm improves cardiometabolic health, as many metabolic and hormonal rhythms peak in the morning, making this an optimal time for food intake [71]. This is supported by a study on humans comparing early TRE to late TRE with the same food intake [65]. The results show many circadian clock genes were expressed in greater amounts in those who practiced early TRE [65]. These results suggest the positive impact eating in sync with the circadian rhythm may have on those with T2D, as the circadian clock is responsible for the natural metabolic processes the body undergoes [25]. However, early TRE may not be effective at reducing glucose levels [66]. Therefore, some studies suggest that early TRE and TRE with small eating windows may provide more benefits to humans with T2D when compared to late TRE and TRE with large eating windows. Conversely, others doubt the effectiveness of these eating patterns on those with T2D.

### 4.2. Alternate-Day Fasting

ADF involves a long, 24 h fasting period. A recent rodent study suggests that ADF reduces glycolipid metabolism disorders through effects on fibroblast growth factor 21 (FGF21) and bile acid metabolism pathways [72]. This study resulted in increased FGF21 hormone production, which is responsible for maintaining glucose, lipid, and energy homeostasis as well as regulating mitochondrial biosynthesis and enhancing energy metabolism [72]. The increased FGF21 expression also improved bile acid synthesis by regulating key genes involved in the process [72]. As a result, ADF promotes bile acid-induced insulin signaling, which regulates glycogen synthesis [72]. Therefore, this fasting method could reduce T2D due to its effects on glycogen synthesis and lipid metabolism. However, a study on adults with obesity reports that ADF did not improve insulin resistance or glucose metabolism, and ADF had results similar to that of caloric restriction in terms of change in weight and lipids [73]. In another study on people participating in ADF, its effect on the future onset of eating disorders such as binge eating and bulimia nervosa were studied [74]. The results suggest that fasting may be a predictor of the future onset of these eating disorders [74]. In conclusion, while some studies point toward ADF providing metabolic benefits in those with T2D, others do not.

### 4.3. 5:2 Fasting

In 5:2 fasting, caloric intake is restricted for two nonconsecutive days of the week. The results of a human study comparing continuous energy restriction and 5:2 fasting suggest that 5:2 fasting may have greater improvements in fasting blood glucose and beneficial changes in appetite ratings [75]. However, the methods have similar outcomes in terms of improving postprandial insulin and glucose responses [75]. These results were similar to those of another study on 5:2 fasting [76]. This study showed that people who fasted had results similar to those of the non-fasting group in terms of changes in fat mass, muscle mass, and percentage of body fat [76]. However, the fasting group did experience a larger difference in body weight [76]. These studies suggest that 5:2 fasting may not be very beneficial for those with T2D.

### 4.4. Long-Term Fasting

While some studies suggest that long-term fasting can provide metabolic benefits that may contribute to a reduction in T2D prevalence, many show concern surrounding the negative effects of long-term fasting. For example, Buchinger fasting, a form of long-term fasting lasting from 4–21 days, has been shown to both improve and worsen certain parameters [77]. This was seen in a study on humans that resulted in an increase in well-being for the subjects [77]. Using this method, fasting resulted in weight loss, reductions in abdominal circumference, reductions in blood pressure, a decrease in blood glucose, and an increase in ketone bodies. [77]. However, serum glutamic oxaloacetic transaminase, serum glutamate pyruvate transaminase, biomarker C-reactive proteins, creatinine, and uric acid all increased [77]. Furthermore, long-term fasting stimulates a stress response, as seen in a human study, which interacts with central serotonin release and may play a role in mood enhancement [78]. A recent case report discusses the advantages and dangers of utilizing a ketogenic diet alongside long-term fasting in those with T2D, as the patient had significant weight loss and better glycemic control [79]. It is also cautioned that fasting along with a ketogenic diet could cause a patient to enter ketosis faster [79]. However, fasting while on a ketogenic diet may cause metabolic acidosis in people with T2D [79]. A recent study on mice tested the prolonged metabolic and epigenetic effects of long-term fasting [80]. This study found that this fasting method causes long-lasting consequences in metabolism, as it makes modifications that alter the body’s response to the next fasting episode [80]. Fasting leads to changes in the hypothalamic expression of transcripts that regulate energy balance and epigenetic modifications, as well as reducing histone acetylation in the neurons of the ventromedial nucleus of the hypothalamus, which is not reversed during refeeding [80]. Additionally, cumulative fasting episodes increase energy intake and reduce energy expenditure after the fasting period [80]. As a result, long-term fasting causes harmful, prolonged effects that may in some cases increase T2D risk. Moreover, long-term fasting increases adipose tissue macrophage levels, as seen in a study on humans, which increases metabolic inflammation [81]. Another study shows that increased inflammation is harmful for people with T2D, as inflammation of adipose tissue is strongly associated with insulin resistance in the obese population [82]. Further, in a study on patients who participated in long-term fasting after gastrointestinal tract surgery, the gall bladder was found to contain sludge, which is believed to evolve into gall stones [83]. Therefore, long-term fasting may cause gallstones by inducing hypotonicity [83]. Some are concerned with the effect of long-term fasting on muscle function, as dietary intake is reduced for long periods of time. A review discusses that the prevalence of sarcopenia could be increased by diets consisting of inadequate nutrients such as protein, vitamin D, and antioxidants [84]. This could indicate that muscle loss may be increased by low nutrition. However, a study on humans undergoing long-term fasting shows that there may not be changes in skeletal muscle properties when fasting is paired with daily physical activity [85]. Lastly, in a study comparing long-term and intermittent fasting, subjects found long-term fasting more difficult [86]. Because of the long-lasting adaptations that take place during long-term fasting, as well as increased metabolic inflammation, long-term fasting may be associated with more disadvantages than advantages for those with T2D.

### 4.5. The Effects of Fasting on the Gut Microbiome

The benefits of intermittent fasting may in part be due to the increase in gut microbiota diversity and changes in the composition it causes. In a study where people fasted 16 h a day for 30 days, participants showed an increase in microbiota diversity and changes in microbiota ratio [87]. Fasting showed an increase in Lachnospiraceae, which suggests improved ability to ferment mucin, giving a competitive advantage when carbohydrates are not plentiful [87]. Increased Lachnospiraceae promotes butryogenesis, improving metabolic effects [87]. These improvements in metabolism could improve harmful effects associated with T2D. However, there is a report that administration of a candidate bacterium belonging to Lachnospiraceae in obese, hyperglycemic mice increased fasting blood glucose levels, increased liver and mesenteric adipose tissue weights, and decreased both plasma insulin levels and HOMA-beta values [88]. These results indicate that Lachnospiraceae may be involved in T2D [88]. A recent study observed a major decrease in the relative abundance of the Firmicutes, Lachnospiraceae, and Ruminococcaceae and an increase in Bacteroidetes and Proteobacteria when healthy subjects underwent ten-day, periodic, long-term fasting [77,89]. These changes in the microbiome were reversed after three months in healthy subjects [77,89]. However, it is also recognized that unhealthy individuals often have less diverse microbial environments [77,89]. Therefore, the microbiome of unhealthy individuals, for example, those with T2D, may not have the same resilience to long-term fasting [77,89]. These results suggest that intermittent fasting with long fasting windows may cause an increase in the Firmicutes/Bacteroidetes ratio, while long-term fasting causes a decrease in the Firmicutes/Bacteroidetes ratio. While some human studies have suggested that an increase in Lachnospiraceae may be beneficial, other rodent studies suggest that Lachnospiraceae is associated with T2D.

The microbiota of the intestine also has its own circadian fluctuations, as some bacteria exhibit diurnal fluctuations in terms of abundance and activity [44]. Community populations and functions also seem to be driven by the time of eating [44]. TRE may also improve levels of specific bacteria involved in metabolic processes and restore the diurnal dynamics of the ileal microbiome and transcriptome, as seen in a mouse study [90]. A recent study on mice found that alterations of the microbiome composition can improve T2D parameters, suggesting butyrate produced by some intestinal bacteria entrains the liver’s circadian clock [91]. The positive effects of TRE on the circadian rhythm of the microbiome are supported by a study in mice that were fed a high-fat diet and had reduced bacterial oscillations [92]. While undergoing time-restricted feeding of a high-fat diet, the circadian rhythms of intestinal bacteria were slightly restored [92]. Furthermore, TRE of the high-fat diet restored glucose tolerance and reduced obesity, as this eating pattern reduced obesogenic bacteria and increased bacteria that promote healthy metabolism [92]. Therefore, just as the host’s circadian rhythm can affect its microbiota, the microbiota can also affect the host [92]. TRE improves the microbiome’s composition and circadian rhythm, leading to metabolic improvements and reduced T2D.

### 4.6. Fasting vs. Pharmacotherapy and Caloric Restriction

The commonly prescribed medication metformin helps in metabolism and alters gut microbiota [43]. A recent study using mouse models suggests that metformin increases glucagon-like protein-1 secretion, suppresses bile acid levels, induces mucin expression, and changes the Bacteroidetes/Firmicutes ratio [93]. Some of these improvements are similar to the effects of Akkermansia, as they both lower tissue inflammation [43]. Moreover, human studies suggest that butyrate-producing pathogens afford protection against T2D [39]. However, xenobiotics change gut microbiota composition and can lead to dysbiosis [43]. In a recent study, obese people with metabolic syndrome received fecal microbiota transplantation from lean human donors, and it resulted in significant improvements in insulin sensitivity [39]. Therefore, as medications may cause dysbiosis, improvements in the gut bacteria composition, as seen with intermittent fasting, may be a more effective treatment for those with T2D.

Additionally, a recent review discusses the effects of intermittent fasting on those with obesity and T2D as superior to the effects of caloric restriction [94]. While both methods produce weight loss, the improvements seen in intestinal microbiota composition, waist circumference, central fat distribution, ketone levels, hunger levels, stress levels, and lean mass preservation are more prominent in those who undergo intermittent fasting [94]. These factors also reduce cardiovascular risk and inflammatory responses, leading to a reduction in T2D parameters [94]. In contrast, other human studies suggest that TRE and caloric restriction are comparable dietary strategies [95]. This included findings of similar reductions in body fat, visceral fat, blood pressure, glucose levels, and lipid levels in caloric restriction with and without TRE [95]. It was also found that intermittent fasting in obese and overweight women resulted in increased inflammation and increased lipolysis when compared to caloric restriction [96]. These findings suggest that there is not consistent evidence for the superior effect of intermittent fasting types with respect to inflammation and lipid metabolism in comparison to caloric restriction. A review discusses that both intermittent fasting and caloric restriction reduced insulin and visceral fat levels as well as increasing insulin sensitivity, but weight loss was slightly superior in those who underwent caloric restriction [97]. However, this same review also found that intermittent fasting reduced glucose levels, while caloric restriction did not [97]. While there are conflicting findings on the effects of intermittent fasting as a comparable or more beneficial dietary strategy to caloric restriction for those with T2D, TRE may be more amenable than other methods because the focus is put on when to eat rather than what to eat [98]. However, intermittent fasting requires proper education and medical monitoring [86]. While caloric restriction and intermittent fasting are comparable in some ways, intermittent fasting was deemed more enjoyable and produced more metabolic benefits in some cases.

In summary, while intermittent and long-term fasting both initially improve parameters of T2D, certain types of intermittent fasting lead to metabolic improvements due in part to its effect on the circadian system, while long-term fasting results in prolonged metabolic consequences [65,80]. Utilizing fasting without proper education could be dangerous and cause harmful effects [86]. For example, while early TRE provides benefits, late TRE leads to negative effects such as increased fasting glucose levels. Furthermore, shorter eating windows (four to six hours) lead to a larger reduction in fasting insulin, further reducing the incidence of T2D [69]. As a result, fasting could be used to reduce the prevalence of T2D, if it is used correctly.

## 5. Conclusions

This review began by providing insight into the mechanistic properties of T2D, revealing the importance of diet management on T2D treatment. In those with T2D, impaired glucose transporters, intracellular insulin signaling, and glucose phosphorylation may occur within muscle cells, resulting in reduced glucose uptake. This leads to the elevated secretion of free fatty acids, increased ectopic fat accumulation, reduced beta-cell function, impaired intracellular signaling, and reduced insulin receptor activity. It is essential that those with T2D are aware of changes they can make in their diet to stop the continuation of regression seen in T2D. Because of the fat reserve genotype adapted by our ancestors, there is a positive relationship between high eating frequencies and obesity and T2D. The disruption in the circadian rhythm of the host may disrupt gut microbiome composition, contributing to increased insulin resistance and buildup of excess energy seen in T2D and obesity. Eating patterns that reduce T2D include regular meal routines, eating breakfast, and decreasing meal frequency when the effects on diet quality are reduced (Figure 1). Therefore, a consistent meal routine with breakfast and lunch may be the best approach to reducing T2D. Fasting methods such as 5:2 fasting, ADF, and long-term fasting all have very few benefits, and in ADF and long-term fasting, the potential harmful effects outweigh the use of these eating patterns for reducing T2D (Figure 1). While some discrepancy still occurs, benefits for early TRE and TRE with short eating windows are supported by human studies indicating the potential for early TRE with short eating windows to reduce T2D prevalence (Figure 1). In addition, eating patterns that lead to improvements in gut bacterial composition may be an effective treatment for those with T2D (Figure 1). The collective available data, therefore, indicate that some eating patterns can be used to reduce the prevalence of T2D in the human population, which is important information given the individual and societal costs of this increasingly prevalent disease.

## Figures and Tables

**Figure 1 nutrients-15-01762-f001:**
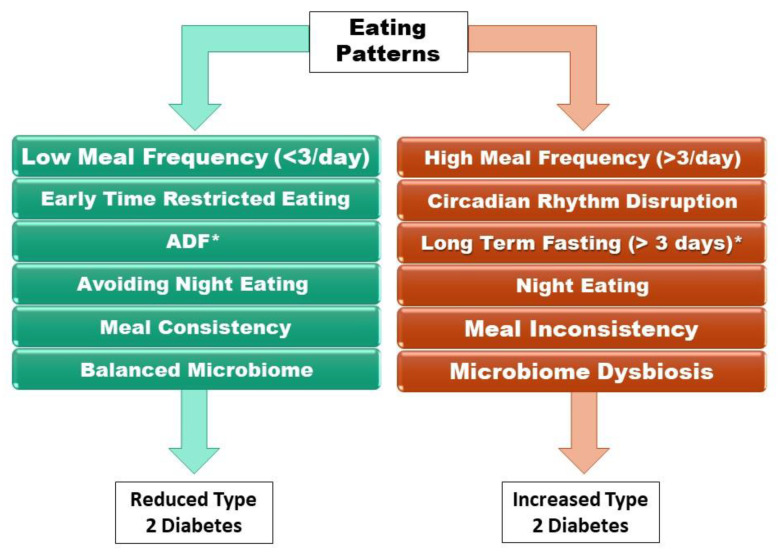
The effects of circadian eating pattern adjustments on T2D. Eating patterns that reduce T2D include lowering the meal frequency to below 3 meals a day and early time-restricted eating. Alternate-day fasting (ADF*) reduces T2D with a risk of eating disorders (bulimia and binge eating). Avoiding night eating, maintaining meal consistency and a balanced microbiome have been shown to reduce T2D. High meal frequency and disruption of the circadian rhythm increase T2D. Long-term fasting without a doctor’s supervision (*), eating at night, meal inconsistency or consuming food that induces gut dysbiosis all increase T2D.

## Data Availability

Not applicable.

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
