# Peer review of "Can Circadian Eating Pattern Adjustments Reduce Risk or Prevent Development of T2D?"

_nutrients, 2023, doi:10.3390/nu15071762_

Round 1

Reviewer 1 Report

Manuscript was written as a review article regarding eating patterns on diabetes risk reduction. The mentioned topic is of clinical importance. In general, I find manuscript well structured regarding subheadings. I suggest improving initial part -introduction- in order to focus main problem discuss through the manuscript. Also, the references should be improved including all the relevant on topic.

Author Response

Cover letter
Journal: Nutrients (ISSN 2072-6643)
Manuscript ID: nutrients-2269768
Title of paper: Can Eating Pattern Adjustments Reduce Risk or Prevent Development of T2D?
Ms. Duska Radanovic
Assistant Editor,
E-Mail: duska.radanovic@mdpi.com
We appreciate the time and efforts by the Editor and Referees in reviewing this manuscript. We 
have addressed all issues indicated in the review report and believe that the revised version meets
the journal’s publication requirements.
Response to comments from Reviewer # 1:
Manuscript was written as a review article regarding eating patterns on diabetes risk reduction. 
The mentioned topic is of clinical importance. In general, I find manuscript well-structured 
regarding subheadings. 
• I suggest improving initial part -introduction- in order to focus main problem, discuss 
through the manuscript. 
Introduction has been improved. It introduces the main idea of the review in light of the 
mechanisms introduced in the introduction in lines 160-163. 
• Also, the references should be improved including all the relevant on topic. 
The references list has been extended to include the most common and relevant studies.
Sincerely,
Dr. Krzysztof Czaja

Reviewer 2 Report

In this review article, the authors described the problems with modern meal frequency and the usefulness of adjusting appropriate eating patterns, such as time-restricted eating, for diabetes and other diseases. The authors' enthusiasm for dietary patterns was strongly evident in this review. However, the prevalence of diabetes is increasing even in countries with large numbers of people observing Ramadan. It has been reported that intermittent fasting improved insulin resistance, but did not improve blood glucose levels (Cell Metab 27: 1212-1221.e1213, 2018). There are reports of intermittent fasting did not improve insulin resistance or glucose metabolism (e.g. Obesity (Silver Spring) 24: 1874-1883, 2016, ). In this review article, there is a mention of Lachnospiraceae, but there is also a report that administration of candidate bacteria belonging to Lachnospiraceae to obese mouse models worsens glucose metabolism (Microbes Environ 29: 427-30, 2014). Thus, the authors should not only state a favorable opinion, but also a paradoxical one. Furthermore, redundancy is the main problem with this review article.

Author Response

Cover letter
Journal: Nutrients (ISSN 2072-6643)
Manuscript ID: nutrients-2269768
Title of paper: Can Eating Pattern Adjustments Reduce Risk or Prevent Development of T2D?
Ms. Duska Radanovic
Assistant Editor,
E-Mail: duska.radanovic@mdpi.com
We appreciate the time and efforts by the Editor and Referees in reviewing this manuscript. We 
have addressed all issues indicated in the review report and believe that the revised version meets
the journal’s publication requirements.
Response to comments from Reviewer # 2:
In this review article, the authors described the problems with modern meal frequency and the 
usefulness of adjusting appropriate eating patterns, such as time-restricted eating, for diabetes 
and other diseases. The authors' enthusiasm for dietary patterns was strongly evident in this 
review. 
• However, the prevalence of diabetes is increasing even in countries with large numbers of 
people observing Ramadan. 
This is explained as Ramadan is an example of late TRE, a fasting method less beneficial than 
early TRE. This is on line 1113-1115.
• It has been reported that intermittent fasting improved insulin resistance, but did not 
improve blood glucose levels (Cell Metab 27: 1212-1221.e1213, 2018). 
This source was added under TRE subchapter, 4.1 on lines 972-1111.
• There are reports of intermittent fasting did not improve insulin resistance or glucose 
metabolism (e.g. Obesity (Silver Spring) 24: 1874-1883, 2016,). 
This is addressed in the alternate day fasting subchapter, 4.2 in lines 1269-1272 and the 
argument made on ADF in the conclusion has been adjusted on line 1586.
• In this review article, there is a mention of Lachnospiraceae, but there is also a report that 
administration of candidate bacteria belonging to Lachnospiraceae to obese mouse 
models worsens glucose metabolism (Microbes Environ 29: 427-30, 2014). 
This has been added in lines 1390-1394. 
• Thus, the authors should not only state a favorable opinion, but also a paradoxical one. 
The argument has been softened throughout. For example, discrepancies between conclusions 
have been addressed in both the introduction and throughout the text. At times this was 
accomplished by adding new references to show contradicting arguments. 
• Furthermore, redundancy is the main problem with this review article.
Many redundancies were removed throughout the text. At times, entire summary paragraphs 
which were not needed were removed. Also, many repeating statements were removed. 
Sincerely,
Dr. Krzysztof Czaja

Reviewer 3 Report

The authors provide a narrative review on dietary strategies to prevent onset and development of T2D.

The overall structure and content of the article is inconclusive in most parts. Rodent studies and human trials are mixed to support the argument of superiority by fasting techniques. The article mixes IF and other fasting concepts without clarification. Review articles are often cited to reference results from single studies. Also, the title does not clarify, that the review is about fasting; "eating patterns" is much more than that. The article needs to be very clear about, what is shown in animals and what is shown in humans. RCTs should be chosen over non-RCTs; SRMAs should be preferred over narrative reviews.

Also, often the term "diabetes symptoms" or "T2DM symptoms" is used. In most cases, this is not correct, as metabolic parameters such as FPG, RR or bodyweight are no symptoms.

In particular, several points should be amended:

Line 25-85, 100-113: The explanation of insulin resistance mechanisms is highly redundant; this section needs serious shortening.

Line 27/28: "occurs" might imply, that T1D does not persist during adulthood. "Onset of T1D occurs" might be better.

Line 32, Line 78: Please clarify, why beta-cell dysfunction (i.e. low insulin output) causes insulin resistance.

Line 38: Association between (total and even saturated) fat intake and T2D is disputed. Please reference accordingly. (https://academic.oup.com/advances/article/13/6/2125/6691423; https://link.springer.com/article/10.1007/s00394-018-1630-4)

Line 51: GLUTs and SGLTs are not defective, but less active / expressed to lower amounts.

Line 70: incretins do not "proteinate" insulin secretion.

Chapter 2.1.:

Generally fine; however, the genetic disadvantage in modern times is redundantly presented.

Chapter 2.2.:
I don't see a good representation of the chapter title in the entire chapter. Hardly any information about modern eating patterns.

Line 205/207: Fully redundant sentence.

Line 216-229: Redundancies.

Line 253-258: redundant to previous sections.

Line 259 ff: The role of leptin is explained quite cloudy. Which effects on leptin are driven by chronic overfeeding, which ones are acute?

Chapter 2.3:
I don't see the point of addressing pancreatic cancer in this article about diabetes prevention.

Chapter 2.4:

Urbanisation is not only irregular meal timing. Also, this chapter is a mixture of negative aspects of urbanisation and of studies improving metabolism by changes of meal timing. In the second part, the writing jumps between circadian rhythm, high fat diet, light and sleep. Once again redundant passages to previous sections.

Also, there is selective argumentation wth cohort studies on meal timing / breakfast skipping. One meal per day is considered a healthy ancient behavior in cohort studies [15], while breakfast skipping (also a reduction in meals) is seen as detrimental urban behavior in the same type of studies. [35,36]

Those contradiction need to be addressed; currently, this chapter is not comprehensive and the conclusion of 2.4 is way too long to actually summarize the content from before.

Chapter 4:

5:2 fasting is not mentioned at all.

Line 412: IF does NOT improve diabetes symptoms. Maybe biomarkers, but not symptoms.

What is the purpose of chapter 4.1, which addresses neither specifically ADF nor TRE and references to only one case report and a animal study? Most of the claims in this paragraph are solely based on rodent studies. Please clarify those intransparencies.

4.2.:

Selective reporting of small single RCTs, even without non-significant differences between IF and control group. Several outcomes related to circadian regulation are currently clinically irrelevant. There is no consistent evidence for a superior metabolic effect of IF with respect to insulin, HbA1c, inflammation, lipid metabolism ... Mixing rodent studies and human trials also masks the absence of clinical evidence.

Compared to 4.3. and 4.4., this chapter looks bloated with redundant information and misleading deductions between preclinical and clinical studies.

4.3.: Not a single reference to a clinical RCT. There is no support by clinical evidence.

4.4.:

Buchinger fasting is not properly referenced by [40].

Reference [41] does not match the content, which it is cited for.

Line 516: This sentence is completely false when compared to the original article, it is based upon.

Line 518 ff: This is not a study, but a case report. It does not show any comparison to other diets. [57]

Well-proven side effects such as eating disorders, gallstones, sarcopenia and others need to be added.

Line 551: [62] does not show prolongevity effects as it is a review article, citing a different article, which includes the mentioned short-term human study, which itself does not show prolongevity.

Line 593: [65] is not a study, but a narrative review, implying superiority. However, all SRMAs do not show that superiority.

Line 606 ff: [43] does not prove superior compliance to IF, as it a three-patient case report.

Conclusion: Way too long, rather a lengthy summary, than a conclusion.

General: The term "diabetics" should not be used.

Author Response

Cover letter

Journal: Nutrients (ISSN 2072-6643)

Manuscript ID: nutrients-2269768

Title of paper: Can Eating Pattern Adjustments Reduce Risk or Prevent Development of T2D?

Ms. Duska Radanovic

Assistant Editor,

E-Mail: duska.radanovic@mdpi.com

We appreciate the time and efforts by the Editor and Referees in reviewing this manuscript. We have addressed all issues indicated in the review report and believe that the revised version meets the journal’s publication requirements.

Response to comments from Reviewer # 3:

The authors provide a narrative review on dietary strategies to prevent onset and development of T2D. The overall structure and content of the article is inconclusive in most parts. Rodent studies and human trials are mixed to support the argument of superiority by fasting techniques.

  • The introduction has been updated to inform the reader that both human and rodent studies are referenced, and some conflicting arguments will be made due to anatomical differences. Furthermore, human studies give superiority over rodent studies throughout the argument. Also, studies are introduced as rodent/rat/mice or human studies.

The article mixes IF and other fasting concepts without clarification.

The Introduction to intermittent fasting was moved to the intro of the chapter so that ADF and TRE are introduced in the beginning of the chapter to prevent nonspecific wording to continue throughout the chapter. Non-specific information on fasting in general has either been identified as one of the categories of fasting or has been removed.

  • Review articles are often cited to reference results from single studies.

The specific single studies related to references were used in the revised manuscript to replace review articles. The new references can be seen in the improved references list.

  • Also, the title does not clarify, that the review is about fasting; "eating patterns" is much more than that.

The conclusion has been updated to show the paper is discussing eating patterns in general, though fasting is included. Most importantly, the final statement of the paper has been adjusted to fit the main idea of the paper. These changes have been made because the review is generally about eating patterns, which is seen in chapters 2 and 3 while fasting is only discussed in chapter 4.

  • The article needs to be very clear about, what is shown in animals and what is shown in humans. RCTs should be chosen over non-RCTs; SRMAs should be preferred over narrative reviews.

Studies have been updated to state if they are from human or animal studies. Many references have been updated to include relevant trials, and many studies which were discussed are now referenced.

  • Also, often the term "diabetes symptoms" or "T2DM symptoms" is used. In most cases, this is not correct, as metabolic parameters such as FPG, RR or bodyweight are no symptoms.

This wording has been addressed. In many cases “symptoms” was changed to “parameters”.

  • Line 25-85, 100-113: The explanation of insulin resistance mechanisms is highly redundant; this section needs serious shortening.

Lines 100-113 were removed to reduce redundancy.

  • Line 27/28: "occurs" might imply, that T1D does not persist during adulthood. "Onset of T1D occurs" might be better.

This change has been made directly.  

  • Line 32, Line 78: Please clarify, why beta-cell dysfunction (i.e. low insulin output) causes insulin resistance.

The wording has been changed to explain that beta-cell dysfunction advances issues seen in glucose uptake deficiencies because it causes abnormal insulin secretion. Low and abnormal insulin output does not directly cause insulin resistance, but it does advance its effects.

  • Line 38: Association between (total and even saturated) fat intake and T2D is disputed. Please reference accordingly. (https://academic.oup.com/advances/article/13/6/2125/6691423; https://link.springer.com/article/10.1007/s00394-018-1630-4)

This source and its results have been added on line 40.

  • Line 51: GLUTs and SGLTs are not defective, but less active / expressed to lower amounts.

This change has been made. “Defective” has been changed to “under-expressed”

  • Line 70: incretins do not "proteinate" insulin secretion.

“Proteinate” has been changed to “increase”.

Chapter 2.1.:

  • Generally fine; however, the genetic disadvantage in modern times is redundantly presented.

A redundant sentence was deleted.

Chapter 2.2.:

  • I don't see a good representation of the chapter title in the entire chapter. Hardly any information about modern eating patterns.

Title was changed to "High vs Low Meal Frequency”.

  • Line 205/207: Fully redundant sentence.

One was removed.

  • Line 216-229: Redundancies.

2 redundant sentences were removed.

  • Line 253-258: redundant to previous sections.

2 sentences previously discussed were removed.

  • Line 259: The role of leptin is explained quite cloudy. Which effects on leptin are driven by chronic overfeeding, which ones are acute?

The main effect, leptin resistance, was stated sooner in the paragraph to emphasize its importance. Also, the concluding sentence was altered to further emphasize the main idea of this paragraph dealing with insulin and leptin resistance.

Chapter 2.3:

  • I don't see the point of addressing pancreatic cancer in this article about diabetes prevention.

This paragraph has been removed.

Chapter 2.4:

  • I believe this section of notes is for 3.4; Urbanisation is not only irregular meal timing.

Both the title and wording within the subchapter have been changed to show that the specific effects of urbanization that deal with T2D are being discussed. 

  • Also, this chapter is a mixture of negative aspects of urbanisation and of studies improving metabolism by changes of meal timing.

These discussions have been separated into 2 separate subchapters. 3.3. The Effects of Urbanization on Type 2 Diabetes, and 3.4. Reducing the Effects of Urbanization.

  • In the second part, the writing jumps between circadian rhythm, high fat diet, light and sleep.

This paragraph was omitted because the content is not directly related to eating patterns, and therefore is making this section seem uncomprehensive.

  • Once again redundant passages to previous sections.

Redundancies were removed.

  • Also, there is selective argumentation with cohort studies on meal timing / breakfast skipping. One meal per day is considered a healthy ancient behavior in cohort studies [15], while breakfast skipping (also a reduction in meals) is seen as detrimental urban behavior in the same type of studies. [35,36]. Those contradiction need to be addressed;

This is now addressed on lines 923-928.

  • Currently, this chapter is not comprehensive and the conclusion of 2.4 is way too long to actually summarize the content from before.

The conclusion has been shortened, and the chapter has been broken up into more subchapters, giving structure. Also, information not regarding eating patterns was removed to allow it to be more comprehensive

  • Chapter 4:
  • 5:2 fasting is not mentioned at all

Chapter 4.3-5:2 fasting has been added on line 1278.

  • Line 412: IF does NOT improve diabetes symptoms. Maybe biomarkers, but not symptoms.

This sentence was removed anyway because it did not address a specific type of IF.

  • What is the purpose of chapter 4.1, which addresses neither specifically ADF nor TRE and references to only one case report and an animal study? Most of the claims in this paragraph are solely based on rodent studies. Please clarify those intransparencies.

This paragraph was removed. The introduction to intermittent fasting was moved to the intro of the chapter so that ADF and TRE are specifically addressed after the introduction.

4.2.:

  • Selective reporting of small single RCTs, even without non-significant differences between IF and control group.

Insignificant results have been addressed on lines 1128-1130, and new studies have been added to this subchapter.

  • Several outcomes related to circadian regulation are currently clinically irrelevant.

Rodent studies were removed from this section to only include relevant effects of TRE on human circadian rhythms.

  • There is no consistent evidence for a superior metabolic effect of IF with respect to insulin, HbA1c, inflammation, lipid metabolism. Mixing rodent studies and human trials also masks the absence of clinical evidence.

The authors revised this part of the manuscript to show that evidence for a superior metabolic effect of IF with respect to insulin, HbA1c, inflammation, and lipid metabolism is still inconsistent. Also, the argument has been softened surrounding the effects of ADFs benefits and early TRE. 5:2 fasting was added and both studies point towards this fasting type having few benefits.

  • Lines 532-544 addresses inflammation and lipid metabolism. Line 365 addresses HbA1c. Insulin is addressed on lines 415-417.
  • Compared to 4.3. and 4.4., this chapter looks bloated with redundant information and misleading deductions between preclinical and clinical studies.

Misleading conclusions have been removed. The argument made in this subchapter is less harsh.

  • 3.: Not a single reference to a clinical RCT. There is no support by clinical evidence.

A human study has been added, and the other information from a rodent trial is presented as such.

  • 4.: Buchinger fasting is not properly referenced by [40].

This has been fixed to “4-21 days.”

  • Reference [41] does not match the content, which it is cited for.

This reference was removed.

  • Line 516: This sentence is completely false when compared to the original article, it is based upon.

Incorrect statements were removed.

  • Line 518 ff: This is not a study, but a case report. It does not show any comparison to other diets. [57]

This is now referenced as a case report and any words indicating comparison have been removed.

  • Well-proven side effects such as eating disorders, gallstones, sarcopenia and others need to be added.

Gallstones were addressed as being a negative side effect of long-term fasting. A study on ADF being a predictor of future onset of eating disorders, binge eating and bulimia, has been added to the ADF section on subchapter 4.2. Sarcopenia and muscle loss due to long term fasting in discussed on lines 461-467.

  • Line 551: [62] does not show prolongevity effects as it is a review article, citing a different article, which includes the mentioned short-term human study, which itself does not show prolongevity.

This was fixed. The correct article is now cited and text does not mention “prolongevity”

  • Line 593: [65] is not a study, but a narrative review, implying superiority. However, all SRMAs do not show that superiority.

This wording was changed to show it is a review.

  • Line 606: [43] does not prove superior compliance to IF, as it a three-patient case report.

The word “prove” has been changed to “may and suggests”, softening the argument here.

  • Conclusion: Way too long, rather a lengthy summary, than a conclusion.

The conclusion has been reduced to include only the main points. All supporting details have been removed.

  • General: The term "diabetics" should not be used.

This term was replaced with “those with T2D”

Sincerely,

Dr. Krzysztof Czaja

Round 2

Reviewer 2 Report

While it is still undeniable that this review article is redundant, I think the authors have revised this review article in accordance with the reviewers' opinions.

Author Response

Cover letter

Journal: Nutrients (ISSN 2072-6643)

Manuscript ID: nutrients-2269768

Title of paper: Can Eating Pattern Adjustments Reduce Risk or Prevent Development of T2D?

Ms. Duska Radanovic

Assistant Editor,

E-Mail: duska.radanovic@mdpi.com

We appreciate the time and efforts by the Editor and Referees in reviewing this manuscript. We have addressed all issues indicated in the review report and believe that the revised version meets the journal’s publication requirements.

Response to comments from Reviewer # 2:

While it is still undeniable that this review article is redundant, I think the authors have revised this review article in accordance with the reviewers' opinions.

Many redundancies were removed, especially in chapter 2.2 and 3.3.

Sincerely,

Dr. Krzysztof Czaja

Reviewer 3 Report

The authors have revised the manuscript in accordance to the reviewer's suggestions.

Some points remain to be addressed:

L. 41: liquid and soft fats (=oils) are not associated with T2D or they are protective.

Chapter 2.2.:

This paragraph requires also the negative impact of meal skipping (breakfast skipping, lunch skipping, dinner skipping). Cohort studies consistently show a detrimental association with T2D and long-term sequelae.

Chapters 2.2. and 3.3 are still highly redundant. Section 3.3. contains a lot of mouse studies, which in the context of urbanisation are hardly suitable sources. The "night eating syndrome" is a defined eating disorder, not induced by poor eating routines, not primarily caused by obesogenic lifestyles.

In conclusion, those two chapters are also very contradicting: Eating one, two, three or four and more meals per day has been individually cited to increase the risk of obesity, T2DM and so on. What's the core message here? The "intermittent" conclusion of l. 307-322 is too lengthy and not concise enough.

Chapter 3.3.:

[45] does not show ANY significant metabolic improvement.

[46] narrative review; cited repeatedly for the same information

Line 390: [52] This is a cross-over RCT; there is no control group besides eTRF and lateTRF.

The conclusion of superiority of lateTRE over earlyTRE is not convincing. Also, there is no indication of benefits of TRE over non-fasting control conditions from the few cited studies. There is even doubt, that TRE has an effect at all.

ADF:

[69] compares hypocaloric ADF to isocaloric ad libitum control. This is no convincing evidence for the specific effect of fasting.

Long-term fasting:

{70} is an observation study without control group. Labelling this study as an evidence for safety of Buchinger fasting, is quite surprising. In all 4 groups, levels of GOT, GPT, CRP, creatinine and uric acid went up.

Line 543: A three-case report cannot indicate any kind of superiority over other therapies.

Conclusion: For neither fasting concept, the review brings up sufficient epidemiological or clinical evidence to claim a beneficial effect of fasting. The only rather consistent source to support fasting are rodent trials.

L. 567: glucose transporters are not defective

Overall, the final chapter exaggerates potential benefits of fasting, which are just not convincingy shown in the previous sections. Cohort studies are inconclusive, TRE, ADF, 5:2 and long-term fasting lack strikingly positive results from (well-controlled) (R)CTs. Fig. 1 ignores these inconsistencies.

The titel of the paper covers "eating patterns", but should speak of "circadian eating patterns", to clarify, that eating patterns defined on food quality are not covered.

The current version of this article is way too enthusiastic towards fasting of any kind. For the final synthesis, even discouraging or warning data are neglected in favor of few positive papers. Thus, the review is flawed by an unbalanced view, based on selective referencing, imprecise citations and cherrypicking desired results.

Author Response

Cover letter

Journal: Nutrients (ISSN 2072-6643)

Manuscript ID: nutrients-2269768

Title of paper: Can Eating Pattern Adjustments Reduce Risk or Prevent Development of T2D?

Ms. Duska Radanovic

Assistant Editor,

E-Mail: duska.radanovic@mdpi.com

We appreciate the time and efforts by the Editor and Referees in reviewing this manuscript. We have addressed all issues indicated in the review report and believe that the revised version meets the journal’s publication requirements.

Response to comments from Reviewer # 3:

Some points remain to be addressed:

  • 41: liquid and soft fats (=oils) are not associated with T2D or they are protective.

This statement has been removed.

Chapter 2.2.:

  • This paragraph requires also the negative impact of meal skipping (breakfast skipping, lunch skipping, dinner skipping). Cohort studies consistently show a detrimental association with T2D and long-term sequelae.

The long-term effects of meal skipping on diet quality are now addressed on lines 180-187, and the effects of skipping breakfast on reducing insulin sensitivity was moved to this paragraph on lines 187-191.

  • Chapters 2.2. and 3.3 are still highly redundant.

Many redundancies were removed.

  • Section 3.3. contains a lot of mouse studies, which in the context of urbanisation are hardly suitable sources.

A human study was added to introduce the topic of irregular eating leading to increased T2D risk on lines 317-319. Animal studies are now only used to propose what may be the cause of this relationship.

  • The “night eating syndrome” is a defined eating disorder, not induced by poor eating routines, not primarily caused by obesogenic lifestyles.

This study was removed because it was not relevant to the discussion on poor eating patterns caused by urbanization.

  • In conclusion, those two chapters are also very contradicting: Eating one, two, three or four and more meals per day has been individually cited to increase the risk of obesity, T2DM and so on.

The conclusions of both chapters 2 and 3 have been updated to more clearly give the message. Specifically, the chapter 3 conclusion now includes aspects of chapter 2 to address the contradictions and give a concise message. This message is now also given in the conclusion chapter and figure.

  • What’s the core message here? The "Intermittent” conclusion of l. 307-322 is too lengthy and not concise enough.

The conclusion has been updated and reduced to clarify the message of this chapter.

Chapter 3.3.:

  • [45] does not show ANY significant metabolic improvement.

This sentence was removed.

  • [46] narrative review; cited repeatedly for the same information

This is now only cited twice to remove the repeated information

  • Line 390: [52] This is a cross-over RCT; there is no control group besides eTRF and lateTRF.

This has been changed to directly state only early and late TRE were compared on line 533.

  • The conclusion of superiority of lateTRE over earlyTRE is not convincing.

A meta-analysis of studies comparing early and late TRE has been added to strengthen the conclusion. Also, the wording of the conclusion sentence has been changed only to suggest that in many cases, only some of the parameters were more improved in early TRE than late TRE.

  • Also, there is no indication of benefits of TRE over non-fasting control conditions from the few cited studies. There is even doubt, that TRE has an effect at all.

The conclusion of this subchapter has been adjusted to express that early TRE is better than late TRE in some ways, but the doubt expressed previously in the subchapter is also now addressed in the conclusion on line 541.

ADF:

  • [69] compares hypocaloric ADF to isocaloric ad libitum control. This is no convincing evidence for the specific effect of fasting.

This reference was removed entirely because it is not relevant in the discussion of the effects of ADF, as it compares hypocaloric ADF to isocaloric ad libitum control.

Long-term fasting:

  • {70} is an observation study without control group. Labelling this study as an evidence for safety of Buchinger fasting, is quite surprising. In all 4 groups, levels of GOT, GPT, CRP, creatinine and uric acid went up.

The negative outcomes of this study are now added in lines 585-994. This source is now used as evidence that long-term fasting has both beneficial and negative outcomes.

  • Line 543: A three-case report cannot indicate any kind of superiority over other therapies.

This case report was removed entirely, as its results are not as strong as the results of the other studies mentioned.

  • Conclusion: For neither fasting concept, the review brings up sufficient epidemiological or clinical evidence to claim a beneficial effect of fasting. The only rather consistent source to support fasting are rodent trials.

The conclusion has been adjusted to discuss that only specific types of IF show evidence of benefits in human studies, such as early TRE and TRE with short windows, and even then, there is some discrepancy. It is now addressed that all other forms have negative effects

  • 567: glucose transporters are not defective

“defective’ has been changed to “impaired”.

  • Overall, the final chapter exaggerates potential benefits of fasting, which are just not convincingy shown in the previous sections. Cohort studies are inconclusive, TRE, ADF, 5:2 and long-term fasting lack strikingly positive results from (well-controlled) ®CTs. Fig. 1 ignores these inconsistencies.

Conclusions have been changed to address the negative or neutral results of many studies on fasting, and the figure has been adjusted accordingly.

  • The tite of the paper covers “eating patterns”, but should speak of “circadian eating patterns”, to clarify, that eating patterns defined on food quality are not covered.

The title has been changed to specify circadian eating patterns.

  • The current version of this article is way too enthusiastic towards fasting of any kind. For the final synthesis, even discouraging or warning data are neglected in favor of few positive papers. Thus, the review is flawed by an unbalanced view, based on selective referencing, imprecise citations and cherrypicking desired results.

Heavily adjusting the conclusions drawn from the data already present in the review to better represent the results and negative outcomes gave a more balanced view on the updated manuscript. Also, some new references have been added, and others lacking validity were removed. The conclusions have been adjusted throughout the paper to more specifically address the negative aspects found in studies on both fasting and other eating patterns. This can be seen in concluding statements at the end of many paragraphs, conclusions of chapters, the conclusion chapter, and the figure.

Sincerely,

Dr. Krzysztof Czaja
